# Analogy in the Civil Law Assessment of Co-Working Agreements in Russia

**Viktor A. Mikryukov**

Department of Entrepreneurial and Corporate Law, Moscow State Law University Named after Kutafin (MSAL), Bldg. 9, Sadovaya-Kudrinskaya Str., 125993 Moscow, Russia; mikryukov.viktor@yandex.ru

**Abstract:** The study's relevance stems from the wider and more active use of shared workstations and the increasing demand for an adequate civil law assessment of contractual co-working relations. The research goal was to identify and, using the analogy, evaluate possible models of legal assessment of co-working contracts from the perspective of civil law. The research methods include special technical and legal tools, such as legal modeling, doctrinal civil means of analysis and synthesis, induction and deduction, and generalization. The research identified the theoretical and law enforcement ambiguity in the legal assessment of co-working contracts and the recommendations on the direct and analogic application of existing civil law instruments to solve this problem. The significance of the study includes its potential to promote co-working as an innovative format for organizing business, work activities, and contributing to an overall increase in the efficiency of civil transactions in Russia. The results of the work may be useful not only for Russian actors, but also for actors in those jurisdictions where co-working also does not have a direct civil law enforcement. At the same time, the research focuses on the potential of analogy in overcoming the ambiguity of the legal regime of new economic phenomena and in creating new legal structures.

**Keywords:** analogy of law; sharing economy; collective office; co-working; unnamedness; contractual structures; legal gaps; lease of workplaces; gratuitous use agreement

## 1. Introduction

The sharing economy implies thatthecollaborative (shared) consumption of goods, works, and services ensures the maximum positive economic effect and low general costs (Botsman and Rogers 2010). This, among other factors, contributes to overcoming crisis phenomena in the economy and ecology (Bogdanova 2020), which underlies the popularity of this economic model not only in the transport industry (carsharing, carpooling, and ridesharing) but also in the use of business (office, retail, or industrial) spaces. The introduction of the sharing economy principles into the possession and use of the non-residential property has led to a new economic phenomenon and a new trend in workstations and entrepreneurship—co-working. The latter, in general terms, means sharing office space and social spaces near the office (Bouncken and Reuschl 2018). In a broad sense, sharing can be anything that refers to pooling resources, products, or services (Matzler et al. 2015).

As in many countries, the number of co-working spaces in Russia has increased. They promote teamwork and cooperation, improving working conditions for self-employed entrepreneurs and employees (Tremblay and Scaillerez 2020; Bychkov 2017). For instance, Russia hosts such well-known network co-working operators as WeWork (https://www.wework.com/ accessed on 20 December 2021), Regus (https://www.myregus.com/, accessed on 20 December 2021), and some developing collective office operators, including Workki (https://workki.co/, accessed on 20 December 2021), Coworkstation (https://coworkstation.ru/, accessed on 20 December 2021), Meeting Point (https://point2meet.ru/, accessed on 20 December 2021), Anyk (http://anyk.biz/kovorking/, accessed on 20 December 2021), and CEO Rooms (https://ceorooms.ru, accessed on 20 December 2021).

The increasing popularity and broader use of co-working as a new economic and social phenomenon (both from the perspective of the organizers of workspaces and their users) necessitate the development of new, adequate approaches to the legal assessment of these emerging relations following their legal specifics and the impact on ancillary and related relationships.

## 2. Literature Review

### 2.1. General Definition of Co-Working

The active digitalization of the economy (Afonasova et al. 2019) and remarkable improvement of information technology contribute to the general increase in the collaborative consumption of goods and services (Hamari et al. 2015). Along with this, against the background of the "uberization" of labor and the growing share of telecommuting, self-employed, and similar types of jobs (Stampfl 2016), the popularity of the co-working economic model is increasing, thus making it a peculiar and significant socio-economic phenomenon (Durante and Turvani 2018).

In Russian civil law, co-working remains an unnamed phenomenon not directly stated in the law. Official documents provide a vague and ambiguous definition of co-working, which is quite inconsistent. For example:

- An equipped space used by visitors for work, communication, and recreation (model standard for the activities of the Center for Cultural Development of Belgorod Region approved by order of the First Deputy Governor of Belgorod Region of 4 August 2015 No. 74); or
- The space for the provision of workstations for temporary use (the procedure for providing state support measures to managing companies of research parks in the field of high technologies, the residents of such research parks, approved by the Resolution of the Government of the Perm Territory No. 533-p of 28 September 2018); or
- Structural subdivision of an organization that is a platform for team and individual activities, a shared workspace for people with different working hours, a co-working space (Order of Rosstat of 28 February 2019 No. 112 "On approval of the federal statistical observation form with instructions for filling it out for the federal statistical observation of the activities of research and development companies by the Ministry of Science and Higher Education of the Russian Federation"); or
- A new labor model that involves sharing rented premises by the self-employed (freelancers), entrepreneurs, and small companies (Law of the Republic of Crimea of 9 January 2017, No. 352-ZRK/2017 "On the strategy of socio-economic development of the Republic of Crimea up to 2030").

The legal purpose of this concept is situational, depending on specific administrative and managerial purposes, which creates unacceptable legal uncertainty regarding the civil legal assessment of co-working relations. Against the background of the general theoretical ambiguity of the subjective or spatial concept of co-working as an economic phenomenon (Uda 2013), this factor necessitates the study of co-working from the civil perspective. Concomitantly, one should check and take into account the potential of analogy as a traditional means of filling legal gaps in private law regulation (Mikryukov 2020).

### 2.2. Benefits of the Co-Working Model for Co-Working Space Users

The main advantage of organizing work in a co-working space is the possibility of acquiring all office amenities flexibly (occasionally varying the range of such amenities by different parameters, choosing certain time intervals by the minute, and receiving related services if necessary) at a relatively low cost (sometimes even free of charge). This economic benefit of the co-working model is critically important for freelancers, the self-employed, aspiring entrepreneurs, and start-ups. In this regard, there is a technical but significant factor that co-working centers may provide an address for a micro-enterprise (Trofimova

2018). The flexibility and low cost of co-working also attract large employers that need out-of-house workstations for the staff doing their work remotely.

Residents of co-working spaces do not have to purchase expensive and burdensome real estate since they may opt for short-term ownership and (or) use, which complies with the consumption trend that has become popular in many countries: not to own things but rent them from the owner for a certain period (Akkermans 2020). Paying for temporary access to a product is often more convenient than permanent possession (Botsman and Rogers 2010), and using the product by receiving access to it rather than owning it prevents one from turning into an enslaved person to the thing (Avdokushin and Kuznetsova 2019).

Access to flexible office space for the self-employed, start-up entrepreneurs and corporations is becoming one of the key resources for business under the conditions of growing external uncertainty (global and regional financial crises, weather disasters, and pandemic consequences). Increasingly, businesses are using scalable space instead of long-term leases or ownership of office space (Gauger et al. 2021).

However, the growing interest in the temporary co-working space is due to direct material savings and the fact that co-working spaces stimulate the exchange of human (labor) and emotional (creative) resources. This exchange allows its residents to compensate for the lack of professional experience, decreases emotional burnout (Ahn 2020), and creates an ecological environment (Artyukhin 2013). This allows the mind to be expanded (stigmatization)—a phenomenon of indirect communication mediated by changes in the environment (Marsh and Onof 2007).

The number of self-employed, freelance, and remote workers is growing steadily. As a result, the increasing number of remote workers start feeling more and more isolated. To address this problem (the problem of social and professional isolation), many independent actors who no longer have to work together choose to do so, at least for some part of their working life (Clifton et al. 2019). From an employer's perspective, the overlap of work and social space can positively affect individual job satisfaction and ultimately increase employees' organizational and innovative potential (Bouncken et al. 2020). In addition, the co-working culture promotes the development of an employee's working skills in a heterogeneous and complexly structured environment (Krause 2019).

Science has persistently considered co-working a highly suitable space for sharing information, which creates a climate that fosters creativity, is open to innovation, and improves the work environment compared to a remote office or company (Tremblay and Scaillerez 2020). Co-working spaces have become a unique phenomenon in the sharing economy that one may regard as a collaborative environment that promotes innovation and creativity with the motto "working together alone" (Berbegal-Mirabent 2021). From this perspective, co-working spaces are even viewed as a kind of "serendipity accelerators" (Moriset 2013).

### 2.3. Advantages of the Co-Working Model for Office Property Owners (Operators)

Economic activity based on the co-working model is beneficial not only for their residents (freelancers, start-ups, remote employees, and other users of co-working spaces), but it is a profitable business or a type of economic activity for the owners of non-residential real estate and those organizing co-working spaces. For instance, according to statistics, it takes three or four months to achieve the operational payback for an average co-working space after its launch, and it takes two years to obtain a full return on investment (Babich and Parkhimenko 2014).

The development of digital technologies (allowing the aggregation of differentiated offers for potential users, remotely informing the latter about the features of a particular real estate object, and concluding agreements on their use in digital format) greatly stimulates the shared use of real estate objects, transforming the mechanism of property transactions and registration of title to it (Emelkina 2021).

From an owner's perspective, the advantage of co-working business models is their additional ability for corporate property management, allowing for the flexible manage-

ment of large corporate assets, ensuring more active and intense use of them (Garrett et al. 2017). The most efficient use of underutilized assets increases the entrepreneur's real estate ownership (this real estate becomes a source of service). At the same time, the concept of real estate ownership becomes redundant for the consumer in the sharing economy. This is why large companies with development funds often create their own co-working spaces (Tremblay and Scaillerez 2020). For instance, large banking sector organizations with a wide branch network prefer leasing to own office spaces (including renting co-working spaces and using desk-sharing technology), purchasing only central (head) offices, or creating more capacious and efficient co-working spaces in their own offices. This allows banks to provide their employees with the necessary working space without exceeding the corporate real estate portfolio (Feshchenko 2021).

As we can see, today, the co-working format has become a fashionable trend and a demanded business niche with expanding geography and content.

### 2.4. The Benefits of Co-Working Relationships for Society and Public Authorities

The public has a positive perception of co-working as it promotes start-ups useful to society and stimulates small and medium-sized businesses, including those launched by the youth (Gazetov 2018). Co-working solves the problem of a limited number of qualified specialists in new areas and contributes to developing innovative regional development programs (Ignatieva 2019), along with a generally positive effect on the urban environment, economy, and planning.

In recent years, co-working spaces have not only become the main neo-corporate model of flexible work in crisis and post-crisis urban economies, but they tend to evolve, turning into sustainable organizational structures, actively communicating with the surroundings and combining entrepreneurship with political and social activities (Gandini and Cossu 2021). The combination of working from home and in a co-working space can compensate for the lack of interaction and social capital for home-based entrepreneurs and increase their income (Rodríguez-Modroño 2021).

From the perspective of urbanism, researchers warn against using co-working spaces as "quick fix" tools for urban renewal (Brown 2017). However, they focus on how co-working generally stimulates residents' business activity and labor productivity in the urban environment and positively influences broader urban transformation processes. It is becoming a unique phenomenon affecting the urban and socio-economic structure and contributing to urban regeneration processes at both local and urban levels (Durante and Turvani 2018). Even now, co-working spaces in Russian cities have become such an element of modern infrastructure, which, along with other public spaces, can revitalize a historic building or area (Gimadeeva and Kinosyan 2020). At the same time, the establishment of co-working spaces as another type of workplace in addition to an office or home in the urban environment is actively occurring not only in Moscow and St. Petersburg but also in the medium and small cities of regional centers.

Russian authorities also perceived various advantages of co-working considered above as an innovative way of organizing workstations, the most effective model for using commercial (non-residential) real estate, and one of the tools for solving social problems.

The national and regional authorities consider co-working centers to develop entrepreneurship, production, and innovation. Thus, co-working should be promoted and supported with appropriate budget subsidies. The following documents regulate this activity:

- Order of the Ministry of Economic Development of Russia of 26 March 2021 No. 142 "On the approval of the requirements for the implementation of measures carried out by the constituent entities of the Russian Federation, the budgets of which receive subsidies for state support of small and medium-sized businesses, as well as individuals using the special tax regime "Tax on professional income" in the constituent entities of the Russian Federation, aimed at achieving the goals, indicators, and results of regional projects that ensure the achievement of goals, indicators, and the results of

federal projects that are part of the national project "Small and medium-sized businesses and support for individual entrepreneurship initiatives", and requirements for organizations that form the infrastructure of support for small and medium-sized businesses";

- Decree of the Moscow Government of 11 October 2011, No. 477-PP "On approval of the state program of the city of Moscow" "Economic development and investment attractiveness of the city of Moscow";
- Order of the Ministry of Investments and Innovations of the Moscow Region of 29 May 2017, No. 47-R "On approval of the application form for receiving subsidies from the budget of the Moscow Region to small and medium-sized businesses to create co-working centers in the Moscow Region";
- The decision of the Kazan City Duma of 14 December 2016 No. 2-12 "On the Strategy of the Socio-Economic Development of the Municipal Formation of Kazan until 2030".

Moreover, co-working is perceived as a mechanism for more efficient solutions of non-commercial and socially significant tasks (Order of the Ministry of Education of Russia of 28 February 2019, No. R-16 "On the approval of methodological recommendations on the creation and operation of advanced training centers", Order of the Ministry of Industry and Trade of the Russian Federation of 20 July 2020, No. 2322 "On amendments to the list of competitive Russian products, the use of which is necessary for the implementation of national projects and a comprehensive plan for the modernization and expansion of the main infrastructure until 2024, approved by order of the Ministry of Industry and Trade of Russia No. 2484 of 15 July 2019").

Co-working has a range of economic benefits for society (for the organizer of commercial co-working space) and as an additional tool for replenishing the revenues of the state budget (Letter of the Ministry of Finance of Russia of 31 October 2018 No. 06-04-11/01/78417 "On methodological recommendations to the executive bodies of the constituent entities of the Russian Federation and local self-government bodies, contributing to the increase in the revenue base of the budgets of the subjects of the Russian Federation and municipal formations").

The factors we considered above indicate that co-working has every chance to integrate into the world and Russian economy completely, tightly, and for a long time. Thus, its adequate legal assessment and proper regulation are increasingly relevant issues. Indeed, when market conditions change dramatically, or new technologies require adjusting regulation rules, the government policy must evolve and adapt to the new reality (Koopman et al. 2015). Therefore, it seems viable to conclude that co-working in Russia will continue to develop, and the number of co-working operators will increase in 2021–2022, along with a greater demand for flexible jobs from corporate clients and, in general, a considerable (possibly twofold) increase in the total area of co-working spaces (See: https://openspace.today/ru/wiki/coworking-prospects-for-the-coming-years/ accessed on 20 December 2021).

*2.5. The Civil Law Assessment of the Relationships between Operators and Residents of Co-Working Spaces*

Researchers have thoroughly studied co-working as an economic phenomenon that evolved and became the focus of attention due to global financial crises and the development of digital technologies.

However, when attempting to assess the manifestations of digitalization and its processes (including the associated trend for the collaborative consumption of goods, works, and services) in modern public relations, experts tend to use economic tools to analyze legal aspects and ignore the "matrix" of civil law (Vasilevskaya 2020). Consequently, they fail to place the new economic relations within the system of time-tested civil legal forms and structures. Thus, the wealth of the available civil law instruments are not being used effectively.

The ongoing transition from possession by ownership to the possibility of prompt access to the property (including modern digital technologies) raises a reasonable question of whether the current legislation can reflect the transition from individual ownership to shared ownership (Mak 2018). To date, this question has not been answered (at least within the civil law assessment of the co-working model).

Legal certainty is a hallmark of the rule of law and a crucial element for a free market economy (Portuese et al. 2013). Moreover, the psychology of Russian legal (judicial) thinking is characterized by its normativity and the inclination to rely on specific rules when assessing legal phenomena (Semenyuta 2021). Therefore, it is vital to provide the most correct and theoretically complete civil law assessment of co-working.

Thus, it is possible to assume that the lack of a general regulatory qualification of co-working from the standpoint of civil law is a legal gap, to overcome which it is possible and necessary to use such a familiar and convenient tool as the analogy of the law.

### 2.6. Materials and Methods

Considering the active introduction of the international practices of collective consumption into the Russian economy and law and the corresponding development of the Russian analogs of global legal structures and economic models, we reviewed the legal, financial, and economic works of international authors. It should, of course, be clarified that the work is not strictly a comparative legal study but relies on the comparability (comparability) of the doctrinal assessment of co-working as an economic and legal phenomenon in Russia and in a number of foreign countries.

A specific feature of the methodological basis of this study was the reliance on the method of economic analysis of law actively applied by international experts. In addition, we examined the co-working phenomenon with the methods of analysis, synthesis, induction, deduction, generalization, and legal modeling.

The normative basis for the study was the provisions of the Civil Code of the Russian Federation (the Code), considering such contractual types that most fully refer to co-working relations (rental agreement, loan agreement, and paid services agreement). In addition, we examined the provisions of this Code that determine the use of contractual structures characteristic of co-working (mixed contract and unnamed contract).

The sources of empirical material for the study were not only the formalized positions of the highest judicial instances (the Plenum of the Supreme Arbitration Court of the Russian Federation, the Presidium of the Supreme Court of the Russian Federation), but also judicial acts adopted in resolving specific disputes caused by the uncertainty of the civil law qualification of co-working agreements.

## 3. Results

The study provided evidence that co-working is one of the phenomena of the sharing economy that in Russia, like in other countries, have the potential for further development (expansion and wider application) both for those involved in sharing office space (operators of co-working spaces and their clients) and for the society (the state and public interest).

The term "co-working" frequently appears in official regulations and business communication. We applied the method of analogy and established that there are several contractual models (and their possible combinations) for the civil law assessment of the relations between owners (operators) of office spaces and their collective users (co-working residents). The main economic aspect of co-working implies paid or gratuitous provision for temporary possession and use or only for temporary use of a part of real estate (office space), which fits into the model of a rental agreement and (or) an agreement of gratuitous use (loan).

As a rule (although not necessarily), additional economic content of the co-working space is attached to the main one. It is regulated by contracts to provide services, agreements, storage, and other civil law contractual models. These types of contracts determine the legal parameters of the optional component of co-working and cannot act as an inde-

pendent legal form of co-working without a rental agreement and (or) an agreement of gratuitous use (loan).

Co-working participants may use legally recognized models of unnamed contracts (which do not have features of mixed agreements), which nevertheless always implies the formation of relations whose essential features are similar to co-working (rent and (or) free use), and may involve building the relations similar to the considered additional co-working relations.

Civil law gives no general normative assessment of co-working is not a real legal gap. Therefore, according to the general rule, co-working contracts cannot be assessed as unnamed, and there is no need to apply the legal analogy.

## 4. Discussion

The taxonomy of modern co-working spaces reflects a wide range of shared workspaces usually regarded as co-working. However, it covers several different economic interactions between operators and residents of shared offices (Orel and Bennis 2021). For instance, according to the Moscow Investment Agency, which performed the Moscow Flexible Workspaces Market Analysis (See: https://innoagency.ru/analiz-coworkings/files/coworkings.pdf accessed on 20 December 2021), there are following types of flexible workstations that are defined as co-working spaces:

- *Conventional*: their basic feature is providing an office space, which implies the allocation of fixed or non-fixed workstations (the so-called classic co-working) or separate offices (mini-offices), or by a combination of classic co-working and mini-offices;
- *Non-conventional*: they provide workstations in addition to their main function; thus, work (office) zones may be located in cafes, anti-cafes, shopping centers, hotels, or residential premises;
- *Specialized*: they provide workspaces for certain social groups or provide non-office workspaces and (or) special equipment for certain non-office activities, including repair, sewing, and other types of workshops (craft co-working), beauty parlors (beauty co-working and tattoo co-working), or children leisure centers (Trushkova and Kvekveskiri 2020).

Each type of co-working space may have different models for its residents:

- *Unlimited access* to the workspace with a fixed payment for a certain period (day, week, or month), regardless of the real-time of using the workstation (access by subscription);
- *Time-based* (hourly) access to the workspace when the payment amount depends on the time spent at the workspace (on-demand access).

Moreover, having examined the conditions of renting a flexible workspace offered by various co-working operators, we established that, regardless of the type of co-working space (a fixed or non-fixed workstation—a place in a real estate object), a co-working resident typically receives some additional amenities (in a set or separately):

- There is a desk, chair, desk lamp, computer, and other equipment required for office or another type of work;
- Access to the shared kitchen, dining areas, dressing rooms, gyms, or recreation areas;
- Reception, postal, and secretarial services;
- Assistance in solving legal and accounting issues;
- IT support.

Considering all this, we may conclude that in terms of civil law, the co-working space refers to various types of contracts, which can have several contractual structures.

### 4.1. Co-Working, Rental, or Free Use

Since the economics of co-working is based on the fact that the operator provides the resident with some part of real estate (office or industrial) for temporary possession and use (or for use only), so first of all, one should establish whether the co-working space fits into the framework of civil law:

- Rental agreement (property lease), as defined in Article 606 of the Code, if it refers to commercial co-working, and
- Agreement of gratuitous use (loan) normatively enshrined in Article 689 of the Code if office premises are shared free of charge.

If a resident is provided with a fixed (pre-determined) equipped workstation (office or other space), regardless of the co-working format used, there are no doubts that the main economic content of the co-working space fully fits into the specified contractual types (rent and (or) loans).

For instance, in these cases, the object of temporary ownership and (or) use is a part of non-residential premises or building (the workstation area) with its characteristics (coordinates, marks on the plan, signs on structural elements, etc.) and some movable objects (office furniture and equipment, devices, or materials). This fully complies with the requirements outlined in Article 607 of the Code for rental and loan objects. These are objects that do not lose their natural properties in their use (non-consumable things), the data about which are stated in the contract, which allows us to establish which property is to be transferred to the resident as an object of lease (ascertained things). At present, a part of real estate can act as an object of lease or loan as there is a direct indication in Article 689 of the Code, and the rules on leased objects are accordingly applied to objects of gratuitous use. This is currently confirmed in Paragraph 9 of the Resolution of the Plenum of the Supreme Arbitration Court of the Russian Federation of 17 November 2011 No. 73 "On certain issues of the practice of applying the rules of the Code on the rental agreement". The Supreme Court of the Russian Federation highlighted that if the parties signed a document containing a graphic and/or textual description of the part that will be used by the lessee and its description implies the parties have agreed on the subject of the rental agreement, then the rental agreement for part of the object should be considered valid. This is confirmed by the Decree of the Judicial Collegium for Economic Disputes of the Supreme Court of the Russian Federation of 6 February 2018, referring to case No. 305-ES17-16171. Law also assumes that the lease (including that for use without transfer of ownership) of a part (element) of an object corresponds to the economic essence of a rental agreement (Zhevnyak 2017).

In general, Russian courts tend to qualify co-working contracts, by which the parties understand the provision of fixed workstations in an open office space, as rental agreements, namely, a rental agreement for a certain part of non-residential premises where the workstations are located (Resolution of the Arbitration Court of the Far East District of 19 January 2015, No. F03-5830/2014; Appellate ruling of the Moscow City Court of 16 June 2021 No. 33-19530/2021). The provision of paid trade places (a sort of co-trading) is defined similarly (as a lease), according to Resolution of the Presidium of the Supreme Arbitration Court of the Russian Federation of 19 November 2013 No. 8668/13 for case No. A82-3890/2012 and Resolution of the Arbitration Court of the North Caucasian District of 2 December 2020, No. F08-11133/2019. Therefore, this approach can refer to similar gratuitous agreements. There is another confirmation that lease and loan models can be applied to the contractual interaction between operators and residents of co-working spaces. Suppose an entity that does not have sufficient authority to lease a real estate object (or part of it) tries to overcome the lack of authority by providing it for use not as part of the premises but considering the workstation as a special object. In that case, the courts recognize that it refers to leasing a part of the real estate (Definition of the Supreme Arbitration Court of the Russian Federation of 13 June 2012, No. VAS-7136/12).

If a resident receives a non-fixed (not particular) equipped workstation in any co-working format, using a lease and (or) loan model is also possible. Even without prior fixation, in the end, the economic needs of a co-working resident are satisfied through physical use (extraction of useful properties) of a specific (identifiable) part of the co-working space not occupied by other residents and specific accessories (furniture or equipment) attached to this part of the real estate.

Here, we would like to note that in some disputable situations, the courts fail to find agreed conditions regarding the specific location, area, and (or) requirements for the equipment of a non-fixed workstation as an object of temporary use by a resident of co-working (or co-trading). Therefore, they assume rental agreements for such vaguely identified workspaces or trade places were not concluded. An example is an agreement between the Silhouette Company and the entrepreneur. According to it, the entrepreneur was to receive "an equipped place in the building at the address of Kirov St., 30" for use. However, the Company did not provide the entrepreneur with the object of the lease, and the court claimed that the contract had not been concluded and rejected the entrepreneur's claim on the obligation of the lessor to provide the resident with an "equipped place" for specific use (Resolution of the Federal Antimonopoly Service of the East Siberian District of 18 August 2005, No. A58-5177/04-F02-3991/05-S2). In another case, the court considered the requirement to release "workstations equipped with racks, desk, and chairs (twenty office workstations and ten warehouse workstations) located in the city of Ufa, Gubaidullina St., 2, according to the rental agreement and indicated that the subject of the agreement in such wording could not be considered agreed. Therefore, the agreement itself had not been concluded (Resolution of the FAS of the Ural District of 06/09/2008 No. F09-3660/08-S6). The court came to a similar decision—that the rental agreement did not comply with the norms on the leased object and, accordingly, the lessor was not to collect rent under the rental agreement for workstations since the lease object was described in the agreement as "a workstation with an area of five square meters" (Resolution of the Arbitration Court of the Moscow District of 13 December 2017, No. F05-16991/2017).

We believe that such private cases contradict the current explanation of the Plenum of the Supreme Arbitration Court of the Russian Federation. It states that the parties executed the agreement even if the lease in the rental agreement is not properly defined. The parties have no right to challenge this agreement on the grounds related to the inaccurate description of the rented object, including references to its non-conclusion or invalidity (Clause 15 of the Resolution of the Plenum of the Supreme Arbitration Court of the Russian Federation of 17 November 2011 No. 73 (as amended on 25 December 2013) "On certain issues of the practice of applying the rules of the Code on the rental agreement"). Therefore, the fact that a specific workstation is not fixed for this resident of a co-working space does not exclude the specification of property characteristics (composition of the workstation, its location, etc.) that the resident will use. What is more, this does not contradict the models of a rental agreement or a gratuitous agreement of property use.

The problem of "double lease" considered in the legal theory does not prevent the regulation of co-working spaces with non-fixed workstations according to the norms on rental or loan agreements (Zaripov 2020). On the one hand, even if the total number of active residents of the co-working space exceeds the number of available workstations, at any given time, only one resident occupies a workstation. On the other hand, the very conclusion of several rental agreements (or agreements on gratuitous use) for the same object does not contradict the current Russian legislation (Clause 25 of the Review of Judicial Practice of the Supreme Court of the Russian Federation No. 1 (2019), approved by the Presidium of the Supreme Court of the Russian Federation on 24 April 2019).

Thus, one should proceed from the fact that, regardless of the format and allocation of a particular workspace to a particular resident, the provision for temporary possession and use (or only for use) of any part of an office (or trade and industrial space) fits into the existing civil legal framework of rent (if the co-working is paid) or free use (if the co-working is non-commercial). The chosen payment model for the used workstation in a commercial co-working space (payment by subscription or for the actual time of use) does not affect this conclusion and does not require the use of additional contractual structures. There is also no joint leasehold on the user's side (Dirkova 2018) since each resident of the co-working space represents an independent party in their relations with the operator within the framework of their separate contract.

Contrary to the opinion of some legal scholars (Ayusheeva 2019), the fact that the relationship between the operator and the resident is short-term, which is characteristic of co-working, does not prevent its full regulation within the framework of these contractual models. This is because most regulatory provisions (on the preemptive right of lease or the distribution of responsibilities according to the type of co-working facilities) imply relations of any duration. Their discretionary nature enables the parties to adjust the provisions inconvenient for short-term co-working.

*4.2. Co-Working, Paid Provision of Services, Free Provision of Services, and Mixed Contracts*

Identifying and comparing the characteristics of renting a workstation with the constitutive features of service contracts are the most challenging aspects in assessing co-working relations. Considering some cases, the courts did not find the necessary specific characteristics when analyzing agreements on providing workstations for rent. However, they did not assume that the contract had not been concluded at all, but qualified the contractual relations for the provision of temporary use of working space as a paid provision of services (Resolution of the Arbitration Court of the Far Eastern District of 5 August 2019, No. F03-3100/2019 and Decree of the Supreme Court of the Russian Federation of 26 April 2018, No. 307-ES18-4272). In one of the cases, the controversial situation arose due to losses incurred by the co-working operator due to the damage to the provided workstations (here, it seems obvious that if the damage is identified, then the object of damage is determined). Nevertheless, the court relied on the Code's provisions regarding the contract for the provision of services for compensation (Resolution of the Arbitration Court of the Moscow District of 24 March 2020, No. F05-1236/2020).

Similarly, some courts proceed because trade places that do not have special characteristics are not independent objects that could be transferred as an object of lease (Resolution of the Federal Antimonopoly Service of the Volga District of 3 March 2010, No. A65-18040/2009). Therefore, courts tend to qualify such relations as paid services (Resolution of the Federal Antimonopoly Service of the Ural District of 21 June 2009, No. F09-5003/09-S5 and Resolution of the Federal Antimonopoly Service of the Central District of 26 January 2010, No. F10-5949/09).

In the legal theory, there is a view according to which a deal with a co-working center can be considered a lease only if the user receives a particular workstation (Bogatyreva and Bogatyreva 2019). However, non-fixed work (or trade) places do not have specific characteristics that would allow the model of a rental agreement or gratuitous use to be applied. Therefore, the interaction between the operator and the resident of the co-working space regarding such objects should be regulated according to the model of service provision (Shestakova and Yavorskaya 2018).

If we extend this approach to the cases involving gratuitous co-working, we will conclude that a gratuitous use agreement does not determine these relations but refers to the gratuitous provision of services.

One can hardly agree with this view. When the workstation is not specified beforehand, it still means that the resident uses the part of the office space occupied by them. Thus, it does not eliminate the fact that the main economic need of the resident of the co-working space will ultimately be satisfied precisely due to their actions by using the required properties of the real estate (office amenities) rather than the actions of the co-working operator.

It is a different matter when some additional amenities (the ability to use the services of common kitchens, dining areas, dressing rooms, fitness spaces, recreation areas, postal and secretarial services, and assistance in solving legal and accounting issues) are stated in the agreements on the use of a non-fixed workstation. As a result, one might think that the listed additions represent the economic basis of co-working relations.

However, it seems that even such attachment of optional co-working elements does not change the basic contractual type of the co-working agreement. From the perspective of civil law, the arising contractual relationship between the operator and the resident of the

co-working space should be regarded as the conclusion of a mixed agreement (Clause 3 of Article 421 of the Code). It contains elements of a rental (or free use) agreement concerning the use of a workstation (part of office space) and elements of other typified contracts regulating the corresponding additions, including elements of contracts for the provision of services, contracts, or storage). At the same time, depending on the parties' specific needs of co-working relationships, contracts regulating their optional part may take the form of a subscription agreement (a contract carried out on demand).

The complex of mixed economic relationships connected with the use of hotel property and the provision of necessary amenities to hotel visitors is regarded as a single normatively named structure of the agreement on the provision of hotel services (Decree of the Government of the Russian Federation of 18 November 2020 No 1853 "On approval of the Rules for the provision of hotel services in the Russian Federation"). Similarly, co-working should be defined as an independent contractual type of paid service, directly stated in the legal norms.

This refutes the idea expressed by researchers that an agreement on the paid use of a part of an immovable thing is a classic example of unnamed contracts without including elements of other (named) contracts (Ksenofontova 2018; Levushkin and Fedchenko 2014) and that the relations under consideration are chiefly regulated by unnamed agreements, which allows the legal analogy to be applied (Poduzova 2021). In most cases, no analogy is required for the contractual assessment of a co-working space.

Co-working parties can conclude legally recognized unnamed contracts that do not have signs of mixed agreements. Following the freedom of contract principle, operators and residents of co-working spaces can intentionally come to an unnamed agreement that does not fit into any known models and does not represent a combination of them. For example, a specific element of ownership and/or use is not defined in advance or afterward, but the parties deliberately speak about the objects that are not typical for rent or gratuitous use. These may refer to "a zone equipped with all necessary means in which an employee or a group of workers can jointly work and perform production tasks". Such a definition of a workstation can be found in Appendix 3 to the Procedure for quoting workstations in companies of the city of Moscow (approved by the Decree of the Moscow Government of 24 March 1998, No 229). In such cases, as follows from the provisions of paragraph 2 of Article 421 of the Code, the general rule does not imply the application of the legal analogy (Clause 1 of Article 6 of the Code) to personal relations of the parties under such agreements, although such possibility is not excluded. However, when businesses use the term co-working, which is not enshrined in the text of the Code and its derivatives (co-working agreements), this does not allow co-working agreementsto be namedas unnamed agreements requiring the application of the legal analogy.

It may become necessary to use an analogy when the optional content of the co-working relationship is done free of charge. Contracts regulating the relations associated with work performance or providing services free of charge are not included in the existing system of named contracts, but they represent unnamed contracts, which are not provided for by the current civil legislation (Solomina 2019). Therefore, the regulation of relations arising from such optional contractual elements of co-working agreements can and should involve the application of the legal analogy.

## 5. Conclusions

Considering the above, the scientific hypothesis put forward about the role of analogy in the civil law qualification of co-working relationships is not fully confirmed. We may conclude that even though common civil law does not define co-working, there are no serious gaps in the regulation of this phenomenon, which would require the application of civil legislation by analogy. The contractual models and structures available in the current civil legislation, combined with the ability to conclude contracts not provided for by law or other legal acts (unnamed contracts), can ensure a perfectly balanced legal interaction between operators and residents of co-working spaces.

According to their different economic and social forms, one should apply a case-specific approach to regulating the diverse relationships connected with the shared use of goods and services (Soifer 2019). The same refers to a group of heterogeneous relations, which some public administrations label using the generalized term "co-working". In each specific case, one should choose the correct civil law model (or find a combination of suitable models from among the ones available) and apply regulatory parameters adequate to a specific situation.

The core of the legal structure of co-working, which reflects the main economic content of these relations, includes rental agreements and gratuitous contracts. The optional co-working component fits into additional typified contractual models (paid services, contracts, or storage), attached to rental agreements and/or gratuitous contracts (for example, loans) as elements of a mixed contract. These circumstances indicate that, in general, there is no need to use an analogy for the contractual assessment of co-working. The fact that civil law contains numerous models of co-working relationships means that there is no need to legalize the term co-working in civil law regulation. At the same time, it is possible that in the long term, if there is a corresponding social and public need, the analogy will be used as a creative factor in the development of legislation. This will contribute to forming a co-working agreement as a separate named contractual type, for example, by analogy with the agreement on providing hotel services. However, such legalization of co-working agreements may be necessary only if they become more widespread and the number of court conflicts arising from them increases.

**Funding:** This research received no external funding.

**Conflicts of Interest:** The authors declare no conflict of interest.

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
