# Peer review of "Analogy in the Civil Law Assessment of Co-Working Agreements in Russia"

_laws_

Round 1
Reviewer 1 Report
I am grateful for the opportunity to review such an interesting draft. Overall, the manuscript is quite well written and argumented. Its subject-matter is innovative and important for today's realities. However, certain points should be clarified and/or improved:
1) The authors say:
Considering the active introduction of the international practices of collective con- | 266 |
sumption into the Russian economy and law and the corresponding development of the | 267 |
Russian analogs of global legal structures and economic models |
(lines 266-268): But what are those 'international practices' more concretely? And what are those 'Russian analogs' in the world, in the global economy? I suggest the authors give a few (at least, a couple) of names of concrete countries, regional integration formations or whatever else they have in mind, and also argument, why they chose these jurisdictions and not others. Otherwise, it is not clear at all to what the actual Russian co-working and its legal and regulatory framework as well as the socio-economic and ecological advantages (as well as downsides) are compared. This leads me to my second, more general, point.
2) In terms of methodology, the authors mention comparative analysis in law, among others. In general, comparative approaches in law look at how relevant laws vary across countries and how improvements, based on
observed models, may be made. The two main types of analysis thereof are (i) functionalist and (ii) universalist. The former inquires into the functions of laws in compared legal systems of selected jurisdictions, requires comparability and underlines not only the differences but also similarities thereof. The universalist analysis looks at how laws interact and eventually glances at the possibility of their approximation.
However, it is totally unclear in the present version of the manuscript which one of the two the authors prefer and use. Actually, it appears that neither of the two has been used by the authors, since the only jurisdiction that the authors analyzed was the law of the Russian Federation. In the lines 268-269, the authors mention the method of comparative jurisprudence. Then, the complete state-of-the-art and maybe even an annotated bibliography should be added to the analysis, so that the reader could clearly see where the results of the analysis come from. Right now such results look somehow coming out of the blue.
Last but not least, the authors mention several times 'comparison' as one of the methodological tools (see eg. line 8). As a lawyer and legal scholar, I immediately expected a comparative legal study, thus using either of the two above comparative law approaches. Since, after the reading of the manuscript, it appeared that it was actually not the case at all, I suggest the authors clarify from the outset (right in the abstract of the article) as well as elsewhere in the text that the 'comparison' they use is the comparison of literature (a comparative discourse analysis, I suspect?--this must be clarified as well!) and not that of laws of different jurisdictions. That is crucial for such journal as 'Laws' as most of your readership is obviously lawyers.
3) Lastly, it is advisable to clearly say what the current framework of co-working in Russia could teach global actors and/or actors in other jurisdictions: eg., what are its advantages/ downsides; what could be improved and how; what could and what could not be transplanted to other jurisdictions (this would be a purely legal comparative conclusion and recommendation, by the way, which definitely would add to the article a great deal of value)--in sum, what the readers of this article who live and work outside Russia will take home? Otherwise, this article will have value only for those who live and work in Russia, which is quite a limited scientific and practical contribution.
All in all, if the aim of the authors was definitely to discuss ONLY the Russian relevant legal and regulatory framework, as well the outcomes and recommendations only for Russian co-working, the authors should state it in a more obvious way starting from the abstract. This would though make their article's reach and usefulness quite limited. If, however, the authors would like to make a more useful contribution, I suggest that they follow the above recommendations regarding the comparisons, and also be clear what they are comparing with what, how and why.
Author Response
The authors thank the reviewer for a careful analysis of the article.
The reviewer’s remarks and recommendations are constructive, their development helped to improve the article.
- First, the abstract and “Materials, and Methods” clarify that the article is not strictly a comparative legal study. The authors are only talking about the fact that the economic and legal doctrinal assessment of co-working in a number of foreign countries (in which, in particular, network co-working companies operate) and in Russia are quite comparable.
The clarifications made it possible to eliminate the grounds for a fair reproach to the reviewer that, in fact, neither the functionalist nor the universalist type of comparative legal analysis was used in the article.
- Despite the fact that for the civil law identification of co-working agreements in this article, the existing civil law models of exclusively Russian legislation were used (which is directly stated in the title of the work), the conclusions drawn in the work may be useful for actors in other jurisdictions, since in the absence of a direct civil law consolidation, co-working is perceived and described in different foreign doctrinal sources in different ways (both as a service agreement, and as a lease contract, or even as a membership). To emphasize this circumstance, a corresponding clarification is added to the abstract.
Reviewer 2 Report
Article title
1. Judging by the article's content, the author focuses not on the whole co-working as a complex economic phenomenon, but on its civil-law contractual models. Therefore, the article's title needs to be clarified.
Abstract and "Materials and Methods" section
2. The author is recommended to correct the wording of the research aim, since the analogy is presented in the work mainly as the study object rather than as a tool.
3. The article is not exclusively doctrinal, but is based, among other things, on the studying the specific empirical material. Therefore, it is necessary to make an appropriate addition.
4. The author should provide information about the specific sources of empirical data for the study (court decisions) and indicate their application scope.
Section "Literature Review"
5. The author should more clearly outline the scientific hypothesis he/she puts forward.
Section "Discussion" ( Subsection 4.2. Co-Working, Paid Provision of Services, Free Provision of Services, and Mixed Contracts, citation (Poduzova 2021))
6. The author declares a refutation of the idea, expressed in the science, that co-working is regulated mainly by unnamed agreements. Clarification is required as to whether the author disagrees with one specific scholar's opinion (Poduzova, 2021) or whether the author is refuting the established scholarly position.
Section "Conclusion"
7. In this section, it is necessary to reflect the result of scientific testing of the hypothesis.
8. The author's rather categorical denial of the need to legalize the term "co-working" in civil law raises some doubts. It is recommended to specify under what conditions such legalization will become necessary.
Author Response
The authors thank the reviewer for a careful analysis of the article.
The reviewer’s remarks and recommendations are constructive, their development helped to improve the article.
- It is necessary to agree with the reviewer that the idea in the article is about civil law contractual models. Therefore, the title of the article has been clarified.
- The emphasis in the work is not on the use of the analogy method in scientific research, but on the analysis of the effectiveness of the application of the analogy method in the practice of filling legal gaps. Therefore, in view of the non-obviousness of the analogous method of reasoning (as indicated by the reviewer), the reference to analogy as a research method was excluded from the annotation and the section “Materials and methods”.
- The “Materials and Methods” are supplemented with an indication of the availability of empirical data used.
- The article is supplemented with information about specific sources of empirical data for the study.
- In “Literature Review”, the scientific hypothesis put forward by the author is clarified.
- The “Discussion” is supplemented with an indication that the opinion criticized by the author on the regulation of co-working, mainly by unnamed agreements, is not a single one, but the author refutes the established scientific position. Specific works of other scientists who share the criticized approach are indicated.
- The “Conclusion” indicates that the scientific hypothesis put forward has not been fully confirmed.
- The prospects for the legalization of the term “co-working” in civil law have been clarified.